# PLANNING FROM PIXELS USING INVERSE DYNAMICS MODELS

**Keiran Paster**
Department of Computer Science
University of Toronto, Vector Institute
`keirp@cs.toronto.edu`

**Sheila A. McIlraith & Jimmy Ba**
Department of Computer Science
University of Toronto, Vector Institute
`{sheila, jba}@cs.toronto.edu`

## ABSTRACT

Learning task-agnostic dynamics models in high-dimensional observation spaces can be challenging for model-based RL agents. We propose a novel way to learn latent world models by learning to predict sequences of future actions conditioned on task completion. These task-conditioned models adaptively focus modeling capacity on task-relevant dynamics, while simultaneously serving as an effective heuristic for planning with sparse rewards. We evaluate our method on challenging visual goal completion tasks and show a substantial increase in performance compared to prior model-free approaches.

## 1 INTRODUCTION

Deep reinforcement learning has proven to be a powerful and effective framework for solving a diversity of challenging decision-making problems (Silver et al., 2017a; Berner et al., 2019). However these algorithms are typically trained to maximize a single reward function, ignoring information that is not directly relevant to the associated task at hand. This way of learning is in stark contrast to how humans learn (Tenenbaum, 2018). Without being prompted by a specific task, humans can still explore their environment, practice achieving imaginary goals, and in so doing learn about the dynamics of the environment. When subsequently presented with a novel task, humans can utilize this learned knowledge to bootstrap learning — a property we would like our artificial agents to have. In this work, we investigate one way to bridge this gap by learning world models (Ha & Schmidhuber, 2018) that enable the realization of previously unseen tasks.

By modeling the task-agnostic dynamics of an environment, an agent can make predictions about how its own actions may affect the environment state without the need for additional samples from the environment. Prior work has shown that by using powerful function approximators to model environment dynamics, training an agent entirely within its own world models can result in large gains in sample efficiency (Ha & Schmidhuber, 2018). However, learning world models that are both accurate and general has largely remained elusive, with these models experiencing many performance issues in the multi-task setting.

The main reason for poor performance is the so-called *planning horizon dilemma* (Wang et al., 2019): accurately modeling dynamics over a long horizon is necessary to accurately estimate rewards, but performance is often poor when planning over long sequences due to the accumulation of errors. These modeling errors are especially prevalent in high-dimensional observation spaces where loss functions that operate on pixels may focus model capacity on task-irrelevant features (Kaiser et al., 2020). Recent work (Hafner et al., 2020; Schrittwieser et al., 2019) has attempted to side-step these issues by learning a world model in a latent space and propagating gradients over multiple time-steps. While these methods are able to learn accurate world models and achieve high performance on benchmark tasks, their representations are usually trained with task-specific information such as rewards, encouraging the model to focus on tracking task-relevant features but compromising their ability to generalize to new tasks.

In this work, we propose to learn powerful, latent world models that can predict environment dynamics when planning for a distribution of tasks. The main contributions of our paper are three-fold: we propose to learn a latent world model conditioned on a goal; we train our latent representation to model inverse dynamics — sequences of actions that take the agent from one state to another, rather

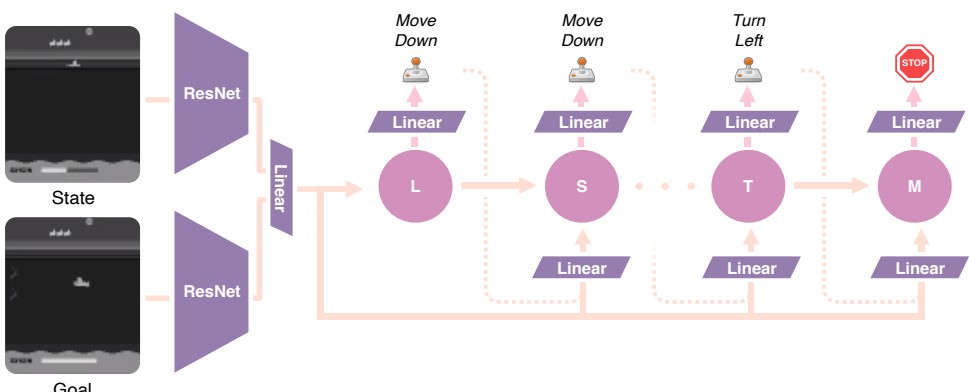

Figure 1: The network architecture for the inverse dynamics model used in GLAMOR. ResNets are used to encode state features and an LSTM predicts the action sequence.

than training it to capture information about reward; and we show that by combining our inverse dynamics model and a prior over action sequences, we can quickly construct plans that maximize the probability of reaching a goal state. We evaluate our world model on a diverse distribution of challenging visual goals in Atari games and the Deepmind Control Suite (Tassa et al., 2018) to assess both its accuracy and sample efficiency. We find that when planning in our latent world model, our agent outperforms prior, model-free methods across most tasks, while providing an order of magnitude better sample efficiency on some tasks.

## 2  RELATED WORK

Model-based RL has typically focused on learning powerful forward dynamics models, which are trained to predict the next state given the current state and action. In works such as (Kaiser et al., 2020), these models are trained to predict the next state in observation space - often by minimizing L2 distance. While the performance of these algorithms in the low data regime is often strong, they can struggle to reach the asymptotic performance of model-free methods (Hafner et al., 2020). An alternative approach is to learn a forward model in a latent space, which may be able to avoid modeling irrelevant features and better optimize for long-term consistency. These latent spaces can be trained to maximize mutual information with the observations (Hafner et al., 2020; 2019) or even task-specific quantities like the reward, value, or policy (Schrittwieser et al., 2019). Using a learned forward model, there are several ways that an agent could create a policy.

While forward dynamics models map a state and action to the next state, an inverse dynamics model maps two subsequent states to an action. Inverse dynamics models have been used in various ways in sequential decision making. In exploration, inverse dynamics serves as a way to learn representations of the controllable aspects of the state (Pathak et al., 2017). In imitation learning, inverse dynamics models can be used to map a sequence of states to the actions needed to imitate the trajectory (Pavse et al., 2019). Christiano et al. (2016) use inverse dynamics models to translate actions taken in a simulated environment to the real world.

Recently, there has been an emergence of work (e.g., Ghosh et al., 2020; Schmidhuber, 2019; Srivastava et al., 2019) highlighting the relationship between imitation learning and reinforcement learning. Specifically, rather than learn to map states and actions to reward, as is typical in reinforcement learning, Srivastava et al. (2019) train a model to predict actions given a state and an outcome, which could be the amount of reward the agent is to collect within a certain amount of time. Ghosh et al. (2020) use a similar idea, predicting actions conditioned on an initial state, a goal state, and the amount of time left to achieve the goal. As explored in Appendix A.1, these methods are perhaps the nearest neighbors to our algorithm.

In our paper, we tackle a visual goal-completion task due to its generality and ability to generate tasks with no domain knowledge. Reinforcement learning with multiple goals has been studied

since Kaelbling (1993). Most agents that are trained to achieve multiple goals are trained with off-policy reinforcement learning combined with a form of hindsight relabeling (Andrychowicz et al., 2017), where trajectories that do not achieve the desired goal are relabeled as a successful trajectory that achieves the goal that was actually reached. Andrychowicz et al. (2017) uses value-based reinforcement learning with a reward based on the euclidean distance between physical objects, which is only possible with access to an object-oriented representation of the state. In environments with high-dimensional observation spaces, goal-achievement rewards are more difficult to design. Nair et al. (2018) use a VAE (Kingma & Welling, 2014) trained on observations to construct a latent space and uses distances in the latent space for a reward. These distances, however, may contain features that are uncontrollable or irrelevant. Warde-Farley et al. (2019) attempt to solve this issue by framing the goal-achievement task as maximizing the mutual information between the goal and achieved state $I(s_g, s_T)$. Our method differs from these approaches since we aim simply to maximize an indicator reward $\mathbb{1}(s_T = s_g)$ and do not explicitly learn a value or Q-function.

## 3 METHOD

### 3.1 PROBLEM FORMULATION

Reinforcement learning is a framework in which an agent acts in an unknown environment and adapts based on its experience. We model the problem with an MDP, defined as the tuple $(S, A, T, R, \gamma)$. $S$ is a set of states; $A$ is a set of actions; the transition probabilities $T : S \times A \times S \to [0, 1]$ defines the probability of the environment transitioning from state $s$ to $s'$ given that the agent acts with action $a$; the reward function $R : S \times A \times S \to \mathbb{R}$ maps a state-action transition to a real number; and $0 \leq \gamma \leq 1$ is the discount factor, which controls how much an agent should prefer rewards sooner rather than later. An agent acts in the MDP with a policy $\pi : S \times A \to [0, 1]$, which determines the probability of the agent taking action $a$ while in state $s$.

The expected return of a policy is denoted:

$$J_{\text{RL}}(\pi) = \mathbb{E}_{\tau \sim P(\tau|\pi)} \left[ \sum_t \gamma^t R(s_t, a_t, s'_t) \right], \tag{1}$$

that is the averaged discounted future rewards for trajectories $\tau = \{(s_t, a_t)\}_{t=1}^T$ of states and actions sampled from the policy. A reinforcement learning agent's objective is to find the optimal policy $\pi^* = \arg\max_\pi J_{\text{RL}}(\pi)$ that maximizes the expected return.

In goal-conditioned reinforcement learning, an agent's objective is to find a policy that maximizes this return over the distribution of goals $g \sim p(g)$ when acting with a policy that is now also conditioned on $g$. In our work, $g \in S$ and we consider goal achievement rewards of the form $R_g(s) = \mathbb{1}(s = g)$. Additionally, we consider a trajectory to be complete when any $R_g(s_t) = 1$ and denote this time-step $t = T$. With these rewards, an optimal goal-achieving agent maximizes:

$$J(\pi) = \mathbb{E}_{g \sim p(g)}[\mathbb{E}_{s_T \sim p(s_T|\pi_g)}[\gamma^T R_g(s_T)]]. \tag{2}$$

Note that unlike prior works, we consider both the probability of goal achievement as well as the length of the trajectory $T$ in our objective.

### 3.2 PLANNING

We consider the problem of finding an optimal action sequence $a_1, \ldots, a_{k-1}$ to maximize expected return $J(s, g, a_1, \ldots, a_{k-1})$:

$$J(s, g, a_1, \ldots, a_{k-1}) = \mathbb{E}_{s_k \sim p(s_k|s, a_1, \ldots, a_{k-1})}[\gamma^k r_g(s_k)] = \gamma^T p(s_k = g|s, a_1, \ldots, a_{k-1}) \tag{3}$$

Thus, the optimal action sequence is found by solving the following optimization problem:

$$a_1^*, \ldots, a_{k-1}^* = \arg\max_{a_1, \ldots, a_{k-1}} \gamma^k p(s_k = g|s_1, a_1, \ldots, a_{k-1}) \tag{4}$$

Even with access to a perfect model of $p(s_k = g|s_1, a_1, \ldots, a_{k-1})$, solving this optimization may be difficult. In many environments, the number of action sequences that reach the goal are vastly outnumbered by the action sequences that do not. Without a heuristic or reward-shaping, there is little hope of solving this problem in a reasonable amount of time.

### 3.3 GLAMOR: Goal-conditioned Latent Action MOdels for RL

Inspired by sequence modeling in NLP, we propose to rewrite Equation 4 in a way that permits factoring across the actions in the action sequence. By factoring, planning in our model can use the heuristic search algorithms that enable sampling high quality language sequences that are hundreds of tokens long. First, note that[1][2]:

$$p(s_k = g | s_1, a_1, \ldots, a_{k-1}) \propto \prod_{i=1}^{k-1} \frac{p(s_k = g | s_1, a_{<i}, a_i)}{p(s_k = g | s_1, a_{<i})} \tag{5}$$

Let $z(s_1, g, a_{<i}, a_i) \triangleq \frac{p(s_k = g | s_1, a_{<i}, a_i)}{p(s_k = g | s_1, a_{<i})}$. Intuitively, these terms are equal to the relative gain in probability of reaching state $g$ conditioned on taking action $a_i$ versus the marginal probability of reaching the goal without conditioning on that action. These terms provide useful information[3] that can guide search towards high scoring action sequences when constructing a plan.

To learn the values of the $z(s_1, g, a_{<i}, a_i)$, we use Bayes' rule to show that we can equivalently learn two auto-regressive behavioral models:

$$\frac{p(s_k = g | s_1, a_{\leq i})}{p(s_k = g | s_1, a_{\leq i-1})} = \frac{p(a_i | s_k = g, s_1, a_{\leq i-1}) p(s_k = g, s_1, a_{\leq i-1}) p(s_1, a_{\leq i-1})}{p(s_1, a_{\leq i}) p(s_k = g, s_1, a_{\leq i-1})} \tag{6}$$

$$= \frac{p(a_i | s_1, s_k = g, a_{\leq i-1})}{p(a_i | s_1, a_{\leq i-1})} \tag{7}$$

We refer to $p(a_1, \ldots, a_k | s_1, s_k = g)$ as the *inverse dynamics model* and $p(a_1, \ldots, a_k | s_1)$ as the *action prior*. Using these models, we can find an optimal plan by optimizing the following objective:

$$a_1^*, \ldots, a_{k-1}^* = \underset{a_1, \ldots, a_{k-1}}{\arg\max} \, \gamma^k \frac{p(a_1, \ldots, a_{k-1} | s_1, s_k = g)}{p(a_1, \ldots, a_{k-1} | s_1)} \tag{8}$$

### 3.4 Learning Inverse Dynamics Model and Action Prior

We parameterize both the inverse dynamics model and the action prior model with LSTMs (Hochreiter & Schmidhuber, 1997) parameterized by $\theta$ and $\phi$. Please refer to section 4.2 for more information about our architecture.

From the training data we construct a new dataset $\mathcal{D}$ of $(s_1, a_1, \ldots, a_{k-1}, s_k)$ tuples for all possible combinations of $s_1$ and $s_k$. Note that this is similar to hindsight relabeling (Andrychowicz et al., 2017). In training, we find parameters to maximize the likelihood of the training data using AdamW (Loshchilov & Hutter, 2019) to optimize the following loss, where $\alpha$ controls the relative weight of the losses when the two models share parameters:

$$L(\theta, \phi) = \mathbb{E}_{(s, a_1, \ldots, a_{k-1}, s_k) \sim \mathcal{D}} [-\log p_\theta(a_1, \ldots, a_{k-1} | s_1, s_k)]$$
$$+ \alpha \cdot \mathbb{E}_{(s, a_1, \ldots, a_{k-1}) \sim \mathcal{D}} [-\log p_\phi(a_1, \ldots, a_{k-1} | s_1)]. \tag{9}$$

While there are many heuristic search algorithms that could be used to optimize Equation 8, we opt to use random shooting (RS) (Richards, 2005) due to ease of parallelization and strong empirical performance. RS samples $N$ candidate action sequences, evaluates the value of the sequences using the learned models, and takes the first action of the winning action sequence. Rather than sampling action sequences uniformly, we sample them auto-regressively using a Boltzmann distribution of the scores $z$.

---

[1]Here we assume that all goals have a non-zero probability of being achieved under the training distribution.

[2]The notation $a_{<i}$ denotes $a_1, \ldots, a_{i-1}$.

[3]By upper-bounding each term $z$, any individual factor being small bounds the rest of the score and allows for early pruning of bad action sequences. An upper-bound can be achieved by assuming some support over all actions in the training policy, which is a common requirement for the convergence of many RL algorithms.

In order to be able to re-plan at every step, our model must be able to model action sequences with variable length. By predicting variable length action sequences using end-tokens, our model is able to plan to both maximize goal-achievement probability and find a shortest path.

---

**Algorithm 1:** GLAMOR

---

Initialize inverse dynamics model $p_\theta(a_1, \ldots, a_{k-1}|s_1, s_k)$;
Initialize action prior model $p_\phi(a_1, \ldots, a_{k-1}|s_1)$;
Initialize dataset $\mathcal{D}((s, a))$;
**for** *training iteration* $k = 1, 2, 3, \ldots$ **do**
    Sample $t$ trajectories with training policy $\pi_k$ and goals sampled from $p(g)$;
    Add tuples $(s_1, a_1, \ldots, a_{k-1}, s_k)$ to dataset $\mathcal{D}$;
    Update $\theta, \phi$ by descending the gradient of Equation 9;
**end**

---

## 3.5 DATA COLLECTION

While training data need not be collected on-policy, we note that using an arbitrary data-collection policy may result in a causally-incorrect model (Rezende et al., 2020). In our experiments, we use our open-loop planner to generate a sequence of actions given just the initial state $s_1$ and a sampled goal $g$ and follow this sequence for the rest of the episode. For a discussion of the types of errors that may occur with a causally incorrect model, see Appendix A.2.

For exploration and in order to ensure support over all actions, we use an $\epsilon$-greedy exploration policy where the $i$th action in the plan is taken with probability $1 - \epsilon$ and an action is randomly sampled with probability $\epsilon$. We decay $\epsilon$ to a small final value over time. We store trajectories in a circular replay buffer, meaning dataset $\mathcal{D}$ contains trajectories from multiple past versions of the planner.

## 3.6 COMPARISON TO PRIOR METHODS

Prior methods for goal-conditioned RL include DISCERN (Warde-Farley et al., 2019) and Goal-Conditioned Supervised Learning (GCSL) (Ghosh et al., 2020). DISCERN learns a goal discriminator and Q-network and rewards the agent for ending up in states where its discriminator can easily guess the desired goal. GCSL learns a policy with supervised learning by iteratively imitating its own relabeled trajectories. While GLAMOR outputs a sequence of actions using an LSTM, these prior works only output a single action for the current time-step.

While GLAMOR shows strong empirical performance in benchmark tasks, we believe that there are several other key advantages over prior goal-conditioned RL algorithms:

- GLAMOR learns a world model, which is more flexible than the Q-networks or policies that prior methods use. For example, despite only being trained to find shortest paths to a goal state, the latent world model learned in GLAMOR contains information about many different paths to the goal. One implication is that during planning, the agent can be modified to select action sequences that reach the goal at a specific time-step. This is experimentally verified in section 4.7.

- Q-learning is well known (Fujimoto et al., 2019) to perform poorly with purely off-policy data and GCSL relies on many iterations of training and data-collection to converge to an optimal policy. As shown in Appendix A.4.1, GLAMOR performs well even with a completely random training policy, often still achieving SOTA performance. Strong off-policy performance may enable the use of offline datasets or more sophisticated exploration policies.

- GCSL does not estimate the prior probability of taking an action sequence and therefore fails to converge to an optimal policy when action sequences that are optimal for one goal may land the agent in another goal state. By using the action prior to disentangle the probability of reaching a goal from the probability of taking an action sequence, GLAMOR avoids this issue. For a more detailed comparison of GLAMOR and GCSL, see section Appendix A.1.

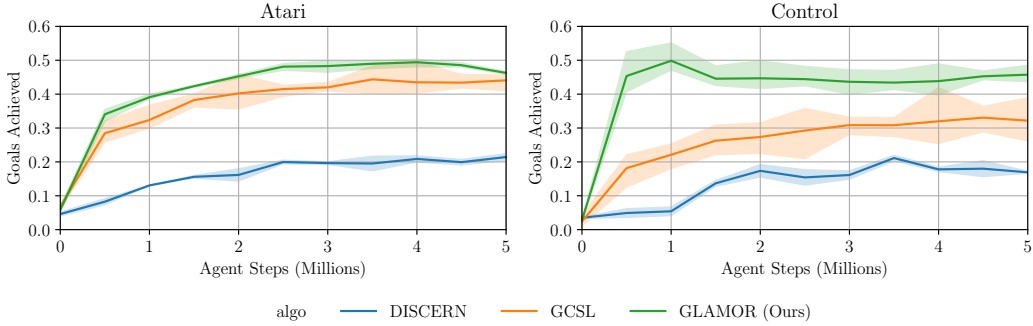

Figure 2: Both in Atari and on tasks from the Deepmind Control Suite, GLAMOR outperforms prior methods. The goal achievement rate is averaged over all games / control tasks and over three seeds. See Figure 9 and Figure 10 in the appendix for more detailed training curves.

## 4 EXPERIMENTS

In our experiments, we aim to answer the following questions:

- How accurate is our world model? Can the agent plan using its model to achieve a diverse set of goals even in high-dimensional domains?
- How sample efficient are our agents? In the low sample regime, is it better to use GLAMOR or use model-free reinforcement learning?
- How effective is our planning procedure? Can the planner discover how to achieve goals with shortest paths or paths with a specific length?

### 4.1 ENVIRONMENTS

We evaluate our method on two types of environments: Atari games and control tasks in the Deepmind Control Suite (Tassa et al., 2018). In both environments, visual observations are converted to grayscale and down-scaled to $(80, 104)$ pixels. In order to incorporate historical information, we opt to concatenate the four most recent frames to form an observation, as introduced by Mnih et al. (2015). However, we opt to use only a single frame for specifying visual goals. We calculate goal achievement in both environments using extracted low dimensional features.

ARCADE LEARNING ENVIRONMENT (ALE). We run experiments on a subset of the available ALE games, chosen by the availability of labeling methods and the suitability of the games to the goal-achieving tasks. In all games we use a frame-skip of $4$, and all Atari environments used a random number of initial noops and sticky actions to introduce stochasticity.

DEEPMIND CONTROL SUITE. We choose to use the subset of the Deepmind Control Suite (Tassa et al., 2018) chosen by Warde-Farley et al. (2019). We also adopt the same method of discretization, discretizing most of the $A$-dimensional continuous control tasks into $3^A$ actions. For `manipulator`, we adopt DISCERN's diagonal discretization. In `point mass` we apply a frame-skip of $4$ frames, with no frame-skip for any other control environment.

Table 1: Goal Achievement Rates at 500k Agent Steps in Atari

| Atari | GLAMOR (Ours) | DISCERN | GCSL |
|---|---|---|---|
| Bowling | 0.14 (0.07) | 0.11 (0.09) | **0.21 (0.11)** |
| Boxing | **0.06 (0.04)** | 0.01 (0.01) | 0.01 (0.02) |
| Breakout | **0.12 (0.05)** | 0.02 (0.02) | 0.04 (0.03) |
| Frostbite | **0.35 (0.06)** | 0.04 (0.04) | 0.20 (0.05) |
| Montezuma | 0.13 (0.07) | 0.03 (0.02) | **0.30 (0.07)** |
| MsPacman | 0.28 (0.07) | 0.04 (0.02) | **0.30 (0.05)** |
| Pitfall | **0.40 (0.08)** | 0.07 (0.04) | 0.26 (0.07) |
| Pong | **0.44 (0.20)** | 0.21 (0.08) | 0.28 (0.06) |
| PrivateEye | **0.26 (0.07)** | 0.04 (0.04) | 0.16 (0.06) |
| Qbert | **0.83 (0.03)** | 0.26 (0.04) | 0.56 (0.17) |
| Riverraid | **0.80 (0.05)** | 0.27 (0.03) | 0.79 (0.15) |
| Seaquest | **0.57 (0.06)** | 0.04 (0.04) | 0.48 (0.08) |
| Skiing | 0.30 (0.09) | 0.02 (0.02) | **0.38 (0.08)** |
| Tennis | **0.10 (0.04)** | 0.00 (0.00) | 0.05 (0.03) |

(a)

| DM Control | GLAMOR (Ours) | DISCERN | GCSL |
|---|---|---|---|
| ball_in_cup | **0.20 (0.08)** | 0.01 (0.02) | 0.03 (0.04) |
| cartpole | **0.97 (0.03)** | 0.01 (0.01) | 0.09 (0.02) |
| finger | **0.09 (0.05)** | 0.01 (0.01) | 0.03 (0.02) |
| manipulator | **0.00 (0.00)** | **0.00 (0.00)** | **0.00 (0.00)** |
| pendulum | **0.83 (0.09)** | 0.20 (0.05) | 0.14 (0.05) |
| point_mass | 0.76 (0.16) | 0.03 (0.02) | **0.82 (0.05)** |
| reacher | **0.27 (0.08)** | 0.08 (0.06) | 0.07 (0.04) |

(b)

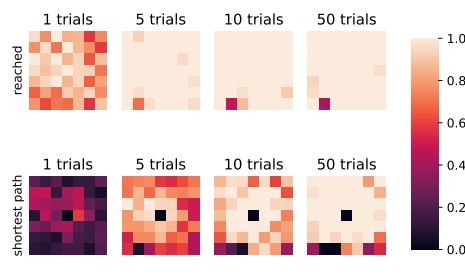

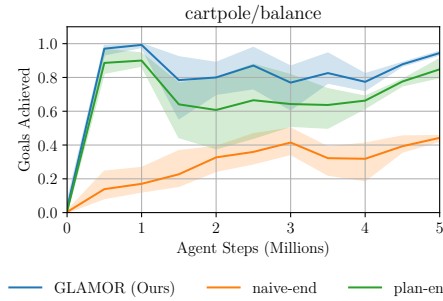

(a) Goal achievement rates for a 7x7 grid-world.

(b) Goal achievement rates for different termination strategies.

Figure 3: (a) The agent starts in the center and must travel to the goal tile. Top shows the rate at which the agent eventually achieved the goal and bottom shows the rate at which the agent achieved the goal with the shortest available path. The amount of compute used for planning is shown on the x-axis. As the planning budget increases, both the number of successfully reached goals and the number of goals achieved optimally improves substantially. Brighter means a higher achievement rate. (b) In "naive-end", the agent greedily tries to take a shortest path to the goal for $T$ timesteps and is evaluated at the end. In "plan-end", the agent explicitly constructs a plan to achieve the goal state at the end of its trajectory. GLAMOR (Ours) can choose to terminate its episode early.

## 4.2 IMPLEMENTATION DETAILS

We parameterize our models with two convolutional encoders, similar to the large model in Espeholt et al. (2018), and an LSTM model (Hochreiter & Schmidhuber, 1997) for action prediction. See Figure 1 for a visual description of our architecture. We opt to share parameters only in the encoders for the inverse dynamics model and action prior. In DISCERN and GCSL, we represent the time remaining with a periodic representation $(\sin(2\pi t/T), \cos(2\pi t/T))$. Detailed training hyperparameters are available at Appendix A.3.

The code for training agents on both Atari and DM Control Suite along with evaluation code can be found at `https://github.com/keirp/glamor`.

## 4.3 BASELINES

We evaluate our method against both GCSL and DISCERN. We chose DISCERN due to its reported high performance on our benchmark tasks, and GCSL due to its similarity to our own algorithm. GCSL was implemented within our code-base, ensuring that any differences are due to algorithmic differences rather than implementation details. Our implementation was checked against publicly available code. For DISCERN, due to the lack of available source code, we made a best effort attempt to reproduce the algorithm. We built our implementation on top of rlpyt's (Stooke & Abbeel, 2019) Rainbow implementation (Hessel et al., 2018). All algorithms use the same encoder architecture, and hyperparameters were fixed between GCSL and GLAMOR.

## 4.4 EVALUATION

To evaluate the degree to which an agent achieved its desired goal, we extract the positions of the various entities in the scene. For Atari, we extract state entity labels using code from Anand et al. (2019). In the Deepmind Control Suite environments, we use the position information from the non-visual representation. Information from both of these agents is kept private from the agents at both train and evaluation time. As in DISCERN, a goal is considered to be achieved if the positions of the entities are within 10% of their feasible range. Positions are evaluated at the end of the episode ($T = 50$ in Atari and $T = 100$ in control tasks) or when the agent decides to terminate the episode. In order to inform GLAMOR of the time remaining until evaluation, we limit the length of the plan in the planner after $t$ steps to $T - t$.

Agents are evaluated on a set of 30 fixed goals per environment. These goals are generated similarly to the *diverse* goal buffer described in Warde-Farley et al. (2019): goals are repeatedly sampled and

a goal is only added to the buffer if it is farther away from the closest goal in feature space than the goal that it is replacing is. We found this procedure to generate a good coverage of possible goals for evaluation.

### 4.5 ACHIEVING VISUALLY-SPECIFIED GOALS

In order to test our model's accuracy, we evaluate our agent's performance in achieving visually-specified goals in all of our test environments. Figure 2 shows that by planning in its latent world model, our agent learns to achieve goals with at least as much accuracy as DISCERN and GCSL in 17 out of 21 tasks, often achieving as much as twice as many goals. See Figure 9 and Figure 10 in the appendix for individual training curves for each environment. Despite not being explicitly trained to only focus on controllable features as in Warde-Farley et al. (2019), Figure 12 shows our agent learning to control the position of its finger even when it can't exactly match the orientation of the spinner. Overall, we conclude that since planning under our agent's world model results in a high goal-achievement rate relative to previous state-of-the-art algorithms, our model is sufficiently accurate. Videos of GLAMOR agents achieving goals in all environments are available at `https://sites.google.com/view/glamor-paper`.

### 4.6 SAMPLE EFFICIENCY

Model-based reinforcement learning is known for having better sample efficiency than model-free algorithms. We found this to be true for GLAMOR as well. In Table 1, we show the performance of our algorithm trained with only 500k agent steps. GLAMOR achieves decidedly more goals than DISCERN and GCSL. Of particular note is that DISCERN is sample inefficient[4], learning to control almost none of the control tasks at 500k steps while GLAMOR has already converged.

### 4.7 PLANNING

We ran several experiments to test properties of our planning procedure.

#### 4.7.1 PLANNING COMPUTE

One benefit of model-based reinforcement learning is that the agent can improve its performance or even adapt its policy at test-time simply by changing its planning procedure. In order to test the performance of GLAMOR with various levels of compute used to construct a plan, we used an empty grid-world environment where shortest-paths are known. We evaluate the percentage of trajectories that reach the goal in an optimal amount of time for each goal. Figure 3a shows a heatmap of the optimal goal-completion rates for different amounts of compute used during planning. As the amount of samples in the planner is increased, both the rate of achieving goals and the rate of achieving goals with an optimal path increase.

#### 4.7.2 EARLY TERMINATION

We also test the flexibility of the learned model by slightly changing the task. While GLAMOR is trained to find shortest paths to the goal state, we ran an additional experiment evaluating its performance if the agent is evaluated on its ability to get to the goal state in exactly $T$ steps. For this experiment, we use the `cartpole/balance` environment, since terminating early when the cart is in the correct position is significantly easier than manipulating it to reach that position at a specific time. In Figure 3b, we show that while early termination achieves the highest goal-achievement rate, explicitly planning to achieve the goal at the end of the trajectory outperforms naively taking shortest paths towards the goal state for $T$ time-steps. All three experiments used the exact same learned model and the only difference is in the planning procedure, where "plan-end" does not sample the termination token until the last time-step. Videos of the different termination strategies are also available on the website.

---

[4]Warde-Farley et al. (2019) trained DISCERN for 200M steps in their paper.

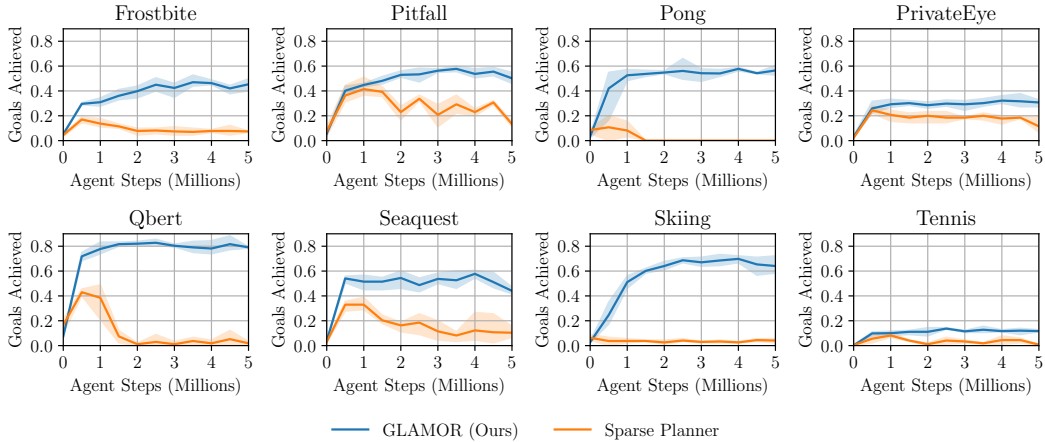

Figure 4: Using intermediate information to guide the planning process helps GLAMOR achieve more goals than when it only looks at the estimated probability of reaching a goal at the end of the episode.

### 4.7.3 SPARSE PLANNING

Prior works such as MuZero (Schrittwieser et al., 2019) and the Predictron (Silver et al., 2017b) also learn latent dynamics models and are similar to GLAMOR in that they predict rewards given a starting state and action sequence. The reward (probability of reaching the goal state) predicted in GLAMOR however is in the form of the inverse dynamics model and action prior, as shown in Equation 8, and the intermediate probabilities predicted by the auto-regressive models are used to guide the search for a good action sequence. To test the contribution of this guidance, we compare GLAMOR to a version where candidate action sequences are generated randomly and the highest scoring one under Equation 8 is selected. Figure 4 shows that performance is significantly worse in the tested environments.

## 5 CONCLUSION

We have presented a novel way to learn latent world models by modeling inverse dynamics. These models learn to track task-relevant dynamics for a diverse distribution of tasks and provide a strong heuristic that enables efficient planning. We also demonstrate strong performance in both the low and high sample regime on 21 challenging visual benchmark tasks.

Goal-achievement tasks already have significant practical value. A next step is to extend GLAMOR to general reward functions. GLAMOR also learns a non-reactive policy. While combining non-reactive planning with Model Predictive Control has proven to be sufficient in many benchmark tasks, a natural future direction is to account for these types of stochastic environments.

### ACKNOWLEDGEMENTS

We gratefully acknowledge funding from the Natural Sciences and Engineering Research Council of Canada (NSERC), the Canada CIFAR AI Chairs Program, and Microsoft Research. Resources used in preparing this research were provided, in part, by the Province of Ontario, the Government of Canada through CIFAR, and companies sponsoring the Vector Institute for Artificial Intelligence (www.vectorinstitute.ai/partners).

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

# A  APPENDIX

## A.1  COMPARISON TO GOAL CONDITIONAL SUPERVISED LEARNING

Ghosh et al. (2020) propose a similar algorithm where a policy is trained by imitation learning with a maximum-likelihood loss:

$$L(\pi) = \mathbb{E}_{\mathcal{D}}[-\log \pi_t(a|s, g, h)] \tag{10}$$

Here $h$ is the time remaining until evaluation. GCSL iterates between training a policy to clone the behavior of the previous policy conditioned on achieving goal $g$ and gathering new training data.

The key differences between GCSL and GLAMOR are: (i) To act, GLAMOR solves an optimization problem while GCSL samples directly from the policy; (ii) GLAMOR learns both an inverse dynamics model and the action prior while GCSL only learns a policy $\pi(s, a, g, h)$; and (iii) Data for GCSL is collected with a reactive policy while GLAMOR uses an open-loop plan.

In our experiments, we found that GCSL performed surprisingly well on visual goal-achievement tasks despite the original implementation by Ghosh et al. (2020) only being tested on control tasks with low-dimensional observations.

Ghosh et al. (2020) provide a theoretical analysis of GCSL, and show that their behavioral cloning objective is a bound on the RL objective, with looseness induced by both how off-policy the training distribution is and the relabeling step. To illustrate the sub-optimal behavior of GCSL caused by relabeling, consider a simplified setting of a one-step MDP and on-policy training data.

**Proposition A.1.** *Let $a_*(g)$ be an action that maximizes the probability $p(s = g|\cdot)$. If there exist two goals $g, g'$ such that $p(g) > 0$, $p(g') > 0$, and $p(s = g|a_*(g)) > p(s = g|a_*(g')) > 0$, then one-step GCSL does not converge to an optimal policy.*

As an example, consider a game with state space $S = \{1, 2, 3, 4, 5, 6\}$ and action space $\{$*roll fair die*, *roll loaded die*$\}$. Rolling the fair die will result in the state being uniformly set to the number that the die lands on, while the loaded die will always land on $1$. The optimal goal-achieving policy is evident: the agent should roll the loaded die with probability one only when the goal is to land on $1$. Otherwise the agent should always roll the fair die. In this example, GCSL will fail to converge but GLAMOR will find the correct policy.

*Proof.* We consider a one-step MDP with goal distribution $p(g)$. In this MDP, we assume a deterministic initial state and denote the state the agent transitions to after taking action $a$ as $s$.

For simplicity, we assume the policy $\pi_\theta$ is powerful enough to perfectly match its training distribution when trained with maximum-likelihood. Therefore, GCSL consists of the following iterated steps:

- Gather trajectories with $\pi_t$. This gives an empirical distribution of goal-conditioned actions $p_{t+1}(a|s = g) = \frac{p(s=g|a)p_t(a)}{p(s=g)}$.

- Distill $p_{t+1}(a|s = g)$ into $\pi_{t+1}(a|g)$.

The update to the policy $\pi$ at each iteration depends both on the relative likelihood of transitioning to the goal with the environment dynamics and the probability of taking action $a$, $p_t(a)$. Clearly, $p_t(a)$ is a function of the previous policy $\pi_t$ and the goal distribution $p(g)$:

$$p_t(a) = \sum_{g'} p(g')\pi_t(a|g') \tag{11}$$

Assuming the distillation is exact, the policy evolves like:

$$\pi_{t+1}(a|s = g) = \frac{p(s = g|a)}{p(s = g)} \sum_{g'} p(g')\pi_t(a|g') \tag{12}$$

We then analyze the ratio of the probability of any sub-optimal action $a$ to the probability of the optimal action $a^*$: $\frac{\pi_t(a|g)}{\pi_t(a^*(g)|g)}$.

If there is *goal interference*, goals $g, g'$ exist such that $p(g')p(s = g|a_*(g')) > 0$ and $p(s = g|a_*(g)) > p(s = g|a_*(g')) > 0$.

Assume $\pi_{t-1}$ is optimal. Then,

$$\frac{\pi_t(a_*(g')|g)}{\pi_t(a_*(g)|g)} = \frac{p(s = g|a_*(g')) \sum_{g''} p(g'')\pi_{t-1}(a_*(g')|g'')}{p(s = g|a_*(g)) \sum_{g''} p(g'')\pi_{t-1}(a_*(g)|g'')} \tag{13}$$

$$\geq \frac{p(s = g|a_*(g'))p(g')}{p(s = g|a_*(g))} \tag{14}$$

$$> 0 \tag{15}$$

Therefore, if $\pi_{t-1}$ is optimal, $\pi_t$ will again be sub-optimal and the policy will never converge. $\qquad\square$

## A.2  CAUSALLY CORRECT MODELS

Rezende et al. (2020) explore the connection between model-based reinforcement learning and causality. A model is *causally correct* if a learned model $q_\theta(x) \approx p(x)$ with respect to a set of interventions. In model-based RL, a model is trained to predict some aspect of an environment. In order to use the learned model to predict the affect of a new policy in the environment, the model must be causally correct with respect to changes in the policy. Rezende et al. (2020) show that some partial models, including MuZero, are not causally correct with respect to action sequence interventions.

| Hyper-parameter | value |
| --- | --- |
| optimizer | AdamW |
| weight-decay | 0.01 |
| normalization | GroupNorm |
| learning-rate | 5e-4 |
| replay-ratio | 4 |
| eps-steps | 3e5 |
| eps-final | 0.1 |
| min-steps-learn | 5e4 |
| buffer size | 1e6 |
| policy trials | 50 |
| state size | 512 |
| clip-p-actions | -3.15 |
| lstm-hidden-dim | 64 |
| lstm-layers | 1 |
| train tasks | 1000 |

Figure 5: Hyperparameters used to train GLAMOR.

As an example, consider training a model to predict whether a sequence of actions wins in the game of Simon Says. The model is trained to predict $p(\text{win}|s_0, a_1, \ldots, a_k)$ on data produced by some training policy. If this training policy is good at the game, the conditional probability $p(\text{win}|s_0, a_1, \ldots, a_k) = 1$ for all action sequences, even though using the action sequence blindly in the real game would result in a much lower win-rate. This happens because the true data-generating process is actually dependent on intermediate states $s_1, \ldots, s_k$, which the training policy has access to. By conditioning on some action sequence $s_0, a_1, \ldots, a_k$ without modeling the intermediate states, these states become confounding variables. In order to predict the affect of taking a certain action sequence on the environment, what we really want to do is find $p(\text{win}|s_0, do(a_1, \ldots, a_k))$.

To learn causally correct models with GLAMOR, we opt to simplify the data-generating process by using training policies that are independent of intermediate states. While this may hurt training in some stochastic environments, we find that in our multi-task setting with relabeling, using non-reactive exploration has a negligible effect.

## A.3  EXPERIMENTAL DETAILS

Figure 5 shows the hyperparameters that were used to train our method. While our method is substantially more simple than value-based methods like DISCERN, there are still a few important hyperparameters. We found that tuning the replay ratio is important to balance between sample efficiency and over-fitting in our models. We also find that GLAMOR works best with a large replay buffer and a large model. We also found that avoiding selecting action sequences which have too low a probability, similar to tricks used in beam search in NLP (Holtzman et al., 2020), increases the performance of our planner. To achieve this, we introduce a hyperparameter `clip-p-actions`, and only expand an action sequence if the immediate log-probability under the inverse dynamics model is over this value.

## A.4  ABLATIONS

In order to find which parts of GLAMOR contribute to its performance, we ran an ablation study.

## A.4.1  TRAINING POLICY

In Figure 6, we vary the way we construct the open-loop action sequence that is followed to collect training data. We vary it in two ways: the amount of compute used to create the plan and whether we consider the action prior. While the performance is poor when using only one planning sample, GLAMOR seems to work well on Pong with as little as 5 samples. More interestingly, disabling the action prior during training seems to increase the variance of GLAMOR significantly. Without the action prior, the planning procedure will select action sequences that may have shown up more in

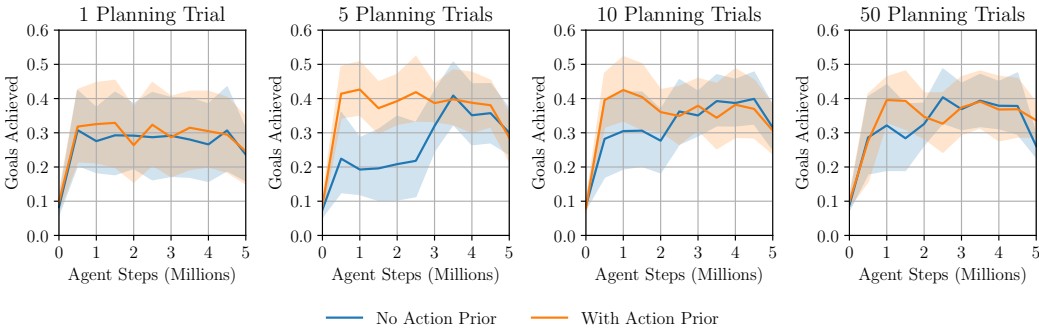

Figure 6: In this experiment, we test how changing the training policy affects performance in Pong. As the amount of compute used in the planner during training increases, so does the performance of the evaluated agent. Using the action prior decreases the variance of the agent's performance.

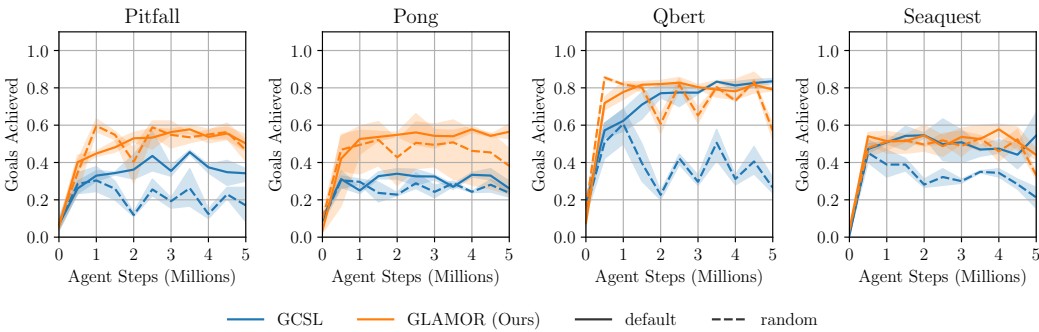

Figure 7: When training agents with off-policy data collected with a random policy, GLAMOR outperfoms GCSL and can achieve most goals.

the training data simply due to the training policy. We hypothesize that this effect can significantly hurt exploration and performance.

We also test the performance of both GLAMOR and GCSL when using a uniform training policy in Figure 7. We found that GLAMOR performs very well even when trained on completely off-policy data, while GCSL struggles.

### A.4.2 PLANNER

We also evaluate whether the planner is necessary at test-time to achieve strong goal-achievement performance. To test this, we ran an experiment where the planner simply takes 1 trajectory sample and takes the first action[5]. Figure 8 shows that while the agent still surprisingly achieves many goals in this setting, using the planner results in a stronger policy. We interperate this result as showing that the heuristic search guided by the factored inverse dynamics and action prior is strong and additional compute simply chooses a plan among already good options.

### A.5 ADDITIONAL FIGURES

In Figure 9 and Figure 10, we plot the learning curves of GLAMOR, GCSL, and DISCERN trained for 5M agent steps. GLAMOR learns quickly compared to the other algorithms and performs better asymptotically with the exception of a few environments (MsPacman, point-mass). Figure 11 and Figure 12 show the states achieved by GLAMOR attemping to reach a specific goal. As noted by Warde-Farley et al. (2019), the `manipulator` environment is difficult and no algorithm learned to

---

[5]Note that this is exactly equivalent to lowering the planning compute in our implementation.

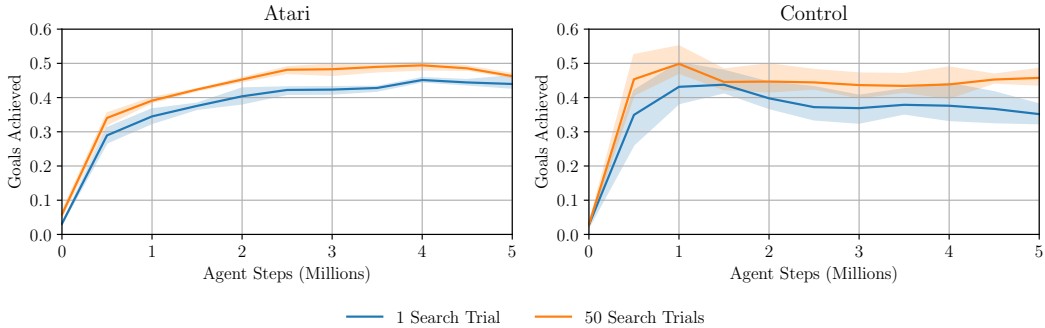

Figure 8: Goal achievement rates for DM Control and Atari. Searching for a high scoring action sequence results in more goals achieved. However, even when using compute equivalent to a model-free agent, GLAMOR performs remarkably well.

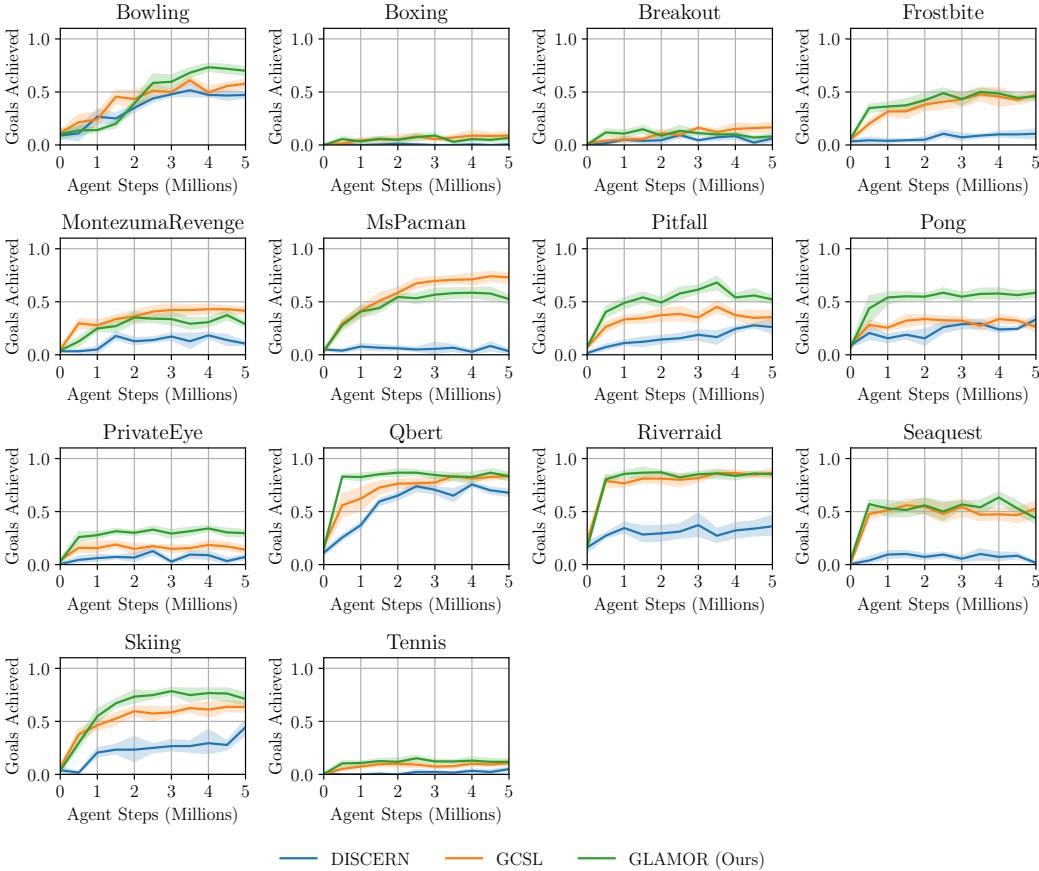

Figure 9: Training Curves for Atari Tasks. GLAMOR achieves more goals (often with many fewer steps) than both GCSL and DISCERN.

achieve goals within 5M steps. The low achievement rate on the `finger` task is due to the agent's inability to reliably control the angle of the spinner.

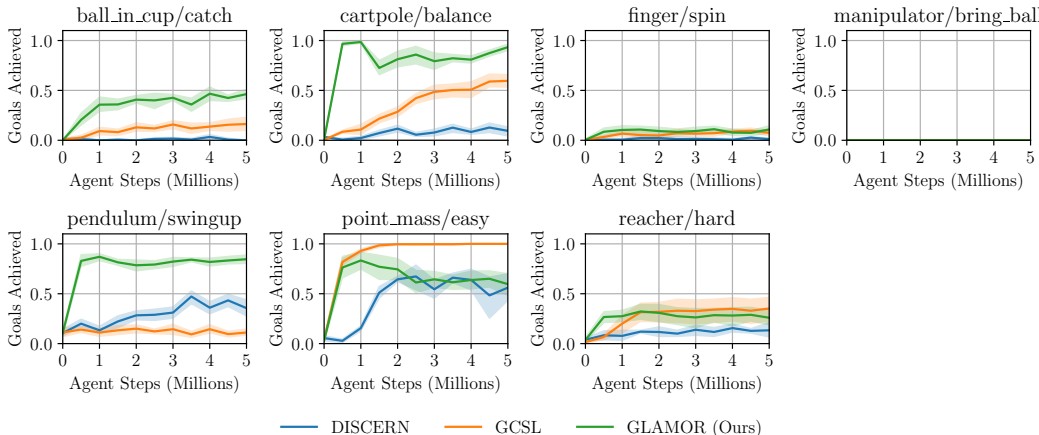

Figure 10: Training Curves for Control Tasks. GLAMOR achieves more goals (often with many fewer steps) than both GCSL and DISCERN.

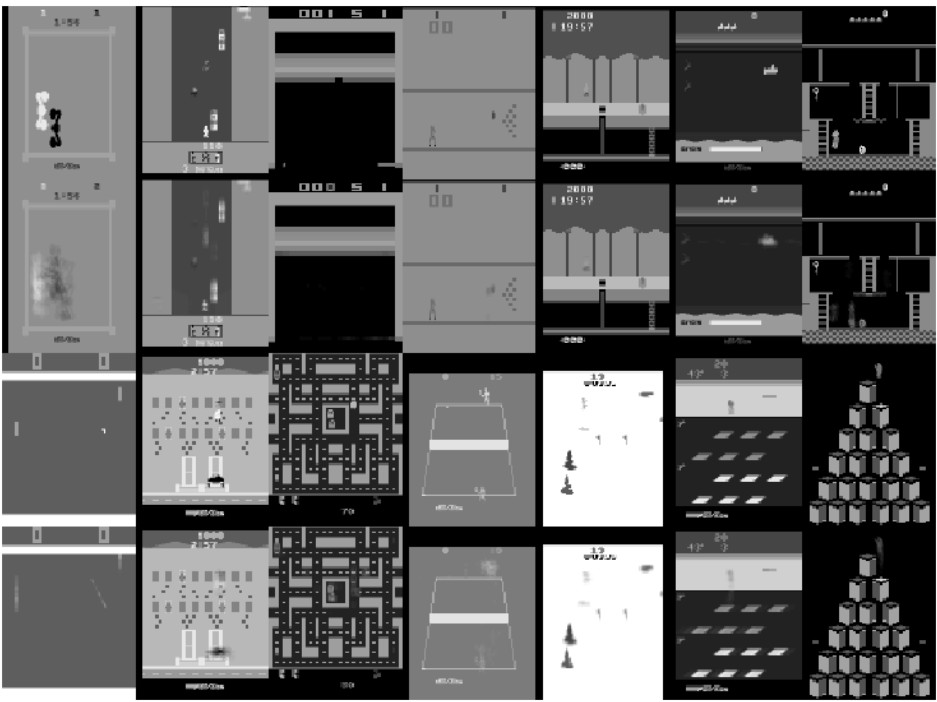

Figure 11: Goal states (above) and states achieved by the fully trained GLAMOR agent (below) averaged over 5 trials for each Atari game tested. Variance comes from environment and planning stochasticity. Note that on most games, GLAMOR learns to control the positions of both directly and indirectly controllable objects in the frame.

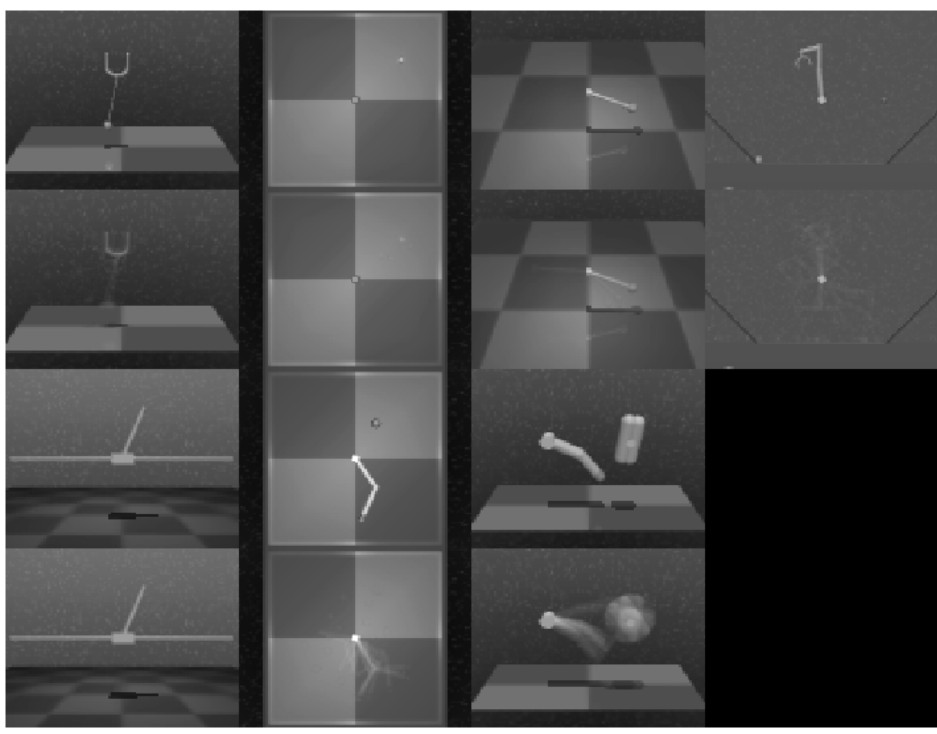

Figure 12: Goal states (above) and states achieved by the fully trained agent (below) averaged over 5 trials for each control task tested. Variance comes from planning stochasticity since the environment dynamics are deterministic. In most environments, GLAMOR learns to control the agent's state to match the visually specified goal. Note that in the finger environment, GLAMOR learns to control the position of the finger despite not often being able to control the angle of the spinner.

