# OpenReview forum: "Planning from Pixels using Inverse Dynamics Models"
_ICLR.cc/2021/Conference — ICLR 2021 Poster_

### Official Review · AnonReviewer2 · 2020-10-28
**This paper tackles the problem of visual goal completion. The authors proposed to learn inverse dynamics (actions instead of states) as the knowledge of the world for planning. They tested their results on several benchmarking tasks and compared to recent methods proposed on goal-conditioned RL and policy learning.**

**Rating:** 6
**Confidence:** 4

**Review:**

The authors adopted an interesting idea for learning inverse dynamics which is different from common approaches where oftentimes a state-transition model is learned for planning. Though in their scenario this does not make any guarantees on the improvement on generalization, conditioning on the rich semantics could be contained in actions, it might be a justification for learning the inverse dynamics model for tasks that test generalization capabilities.

However, the idea of using relabeling to facilitate the learning of goal-conditioned policies and world models is not new as discussed by the authors. This makes the learning framework somewhat less significant in novelty. More specifically, the authors put quite a lot of effort into distinguishing their method from a recent method GCSL. The differences actually lead to a question on the action prior. I'm curious if this prior distribution is different from a uniform distribution over all available paths starting from $s_1$, it is not well justified why we need to parameterize this with LSTM.

There are also several, I believe, typos that might need fixing. The first term in Equation (9), should it be $\mathbb{E}[-\log p_{\theta}(a_1, \cdots, a_k | s_1, s_k)]? In the experiment section, the model was trained with 500k agent steps while in the figure it was 5M, this leads to the problem of sample efficiency (section 4.6).

Overall, I feel that though the idea of adopting inverse dynamics for learning action model is interesting, it is still somewhat incremental to prior works with a similar framework. The missing justification and details also contribute to the decision of putting a decision of borderline during pre-rebuttal period.

================================================================================================================

The authors has addressed my concerns well in the discussion period, I have increased the score to 6.

---

> ### Author Response · Authors · 2020-11-17
> **Thank you for the comments**
>
> Thank you for the valuable comments. We have uploaded a new PDF. Please see general comments to all reviewers for details about what has changed.
>
> Q: The learning framework is not novel.
>
> Our work is the first to our knowledge to use temporally-extended inverse dynamics to learn a latent world model, and the first to show how this model can be combined with an action prior to efficiently plan to control the state of an environment. Although GCSL seems similar since it also trains on its own, relabeled, trajectories, we think we may have overstated the similarities in our original submission. GCSL is fundamentally a model-free RL algorithm which learns a policy to achieve goals while GLAMOR learns a latent dynamics model that predicts entire action sequences.
>
> We updated section (3.6) in the updated PDF to outline some advantages of our approach over the prior works. Additionally, we added some additional experiments that we think empirically show the advantage of GLAMOR over prior works.
>
> Section 4.7.2: GLAMOR’s latent world model is more powerful than a policy. We ran an experiment where, at test time, the agent searches for action sequences that lands it in the goal state at exactly the last time-step in the episode rather than searching for a shortest path. Although GLAMOR is not trained on this particular task, it still achieved impressive performance because the world model knows about many different action sequences that lead the agent to the goal.
>
> Section 4.7.3: One of the main benefits of using an inverse dynamics model is that we can use the factored probabilities to guide the search for a strong action sequence. This is in contrast to how an algorithm like MuZero (Schrittwieser et al., 2019) models the expected reward at each time-step and learns a separate policy/value function to guide search. We added a new experiment to test the extent to which using an inverse dynamics model helps with planning versus directly modeling the reward the agent cares about (probability of reaching the goal) and planning using just sparse information.
>
> We used RS in this experiment as well, sampling action sequences at random and picking the sequence with the highest score out of the sampled sequences. We found that in our environments, this agent performed significantly worse with the same level of compute, showing the value added by learning an inverse dynamics model.
>
> Appendix A.4: We added an ablation study which shows: GLAMOR clearly benefits from using the action prior; planning during training is important, but GLAMOR can get away with using relatively little (5 samples performs almost as well); and GLAMOR performs much better than GCSL on truly off-policy data collected via a random walk.
>
> Q: Is the action prior different from a uniform distribution?
>
> Indeed, if the training policy is uniform then the action prior is very simple. But when we do iterative training or we have a non-uniform training policy, the action prior is a complicated combination of the training policies averaged over the goal states in the dataset. We use an LSTM to learn this distribution. Appendix A.4.1 shows that disabling the action prior when training in Pong results in substantially worse performance compared to the full GLAMOR algorithm.
>
> Q: Equation (9) typo
>
> Fixed in the new draft.
>
> Q: Table shows agents at 500k steps
>
> We show a table of the results at 500k steps because it highlights that our method performs really well even with that low amount of training data. The training curves are for the full 5M steps of training.
>
> Q: Justification and Details:
>
> Please see the appendix for details, including an ablation study that provides the reader with further insight into the effectiveness of the technique.

---

> > ### Comment · AnonReviewer2 · 2020-11-24
> > **Response to the author**
> >
> > Thank you for your clarification and adjustments made to make this paper better. The prior distribution problem makes sense and solved my concern. As for the comparison between the learned inverse dynamic model and commonly-used expected reward prediction, I agree with the authors that sometimes strong action sequences lead to faster and better performance. However, one new concern from that point in the rebuttal is, having a strong action sequence distribution from training experience is not always a good idea, especially when meeting changing environments with similar dynamics (e.g. a simple grid world with traps but in different positions). Though beyond the current paper's scope, to which problems will the proposed method give performance boosts on is also an interesting topic to discuss.

---

### Official Review · AnonReviewer1 · 2020-10-28
**Planning from Pixels using Inverse Dynamics Models**

**Rating:** 6
**Confidence:** 3

**Review:**

The paper proposes to learn task-agnostic world models in latent space, using a goal-conditioned inverse dynamics formulation and an action priors model, that learn to predict action sequences. The approach is compared against a model-free unsupervised control method and an imitation learning method on the subset of Atari games and a continuous control benchmark, and showed to achieve superior performance in most tasks based on goal completion ratio and sample efficiency.

The proposed approach learns two networks, the inverse dynamics model and the action prior model, and uses this to plan a sequence of control actions, rather than learning the policy directly as done by the baselines. Although this has shown to achieve more goals in the 2 benchmarks, the authors do not discuss the learning complexity of training two networks compared to one policy network as done in baselines. The authors conduct an ablation study on the computational complexity of planning at test-time for their approach. I believe it would help to evaluate the method better if they also show the number of trials used when comparing to baselines.

Can the authors provide any real-time performance guarantees at test time?

It's not clear to me how the exploration and exploitation is traded off in the Data Collection section. The planner is used to generate a sequence of actions but an e-greedy policy is used for exploration. This is confusing to me.

The paper claims that the inverse dynamics formulation learns to represent controllabe aspects of the state. Has this been theoretically proven before? And have any empirical studies been done to show that this actually helps model-based RL to perform better than when using forward models? It would be interesting to compare against model-based RL techniques based on forward models, if avaialble, for this work.

---

> ### Author Response · Authors · 2020-11-17
> **Thank you for the comments**
>
> Thank you for the valuable comments. We have uploaded a new PDF. Please see general comments to all reviewers for details about what has changed.
>
> Q: authors do not discuss the learning complexity of training two networks compared to one policy network as done in baselines
>
> Good question. Interestingly, the addition of an additional network has minimal impact in practice. This is because: 1) the bottleneck in the algorithm is collecting data from the environment rather than training the neural networks; and 2) we share parameters between the ResNet used to encode observations in the inverse dynamics model and the action prior, which reduces the number of additional parameters and allows us to only compute it once.
>
> Q: I believe it would help to evaluate the method better if they also show the number of trials used when comparing to baselines.
>
> We have added an extensive ablation study in Appendix A.4. Among other experiments, we show that: the performance of our algorithm is still strong when it is trained even on data collected via a random policy; using training trials as low as 5 still yields strong, state of the art performance; and even when using just one planning sample at test time (equivalent compute to model-free), GLAMOR suffers only a minor hit in performance and still achieves more goals than the baselines.
>
> Q: Can the authors provide any real-time performance guarantees at test time?
>
> GLAMOR needs to search for a strong action sequence while the baseline methods have a policy/q-function that make inference much quicker at test time. However, we find that our planner only takes around 0.1 seconds to plan on a P100 GPU. These plans contain not only the next action, but also the entire sequence of actions that the agent currently thinks would be optimal, meaning that if real-time performance is a concern, re-planning could happen only every few steps and the agent could follow the open-loop plan in between re-planning. Another option is to directly sample actions rather than planning, as in Appendix A.4.2, which we show only slightly reduces performance.
>
> Q: The planner is used to generate a sequence of actions but an e-greedy policy is used for exploration.
>
> We apologize for not being clear about the implementation of this. We have fixed it in the updated paper. At the beginning of each episode, a goal is sampled and the agent generates a sequence of actions. At each step during that episode, the agent takes the ith action in the plan with probability 1-eps and takes an action randomly with probability eps.
>
> Q: The paper claims that the inverse dynamics formulation learns to represent controllable aspects of the state. Has this been theoretically proven before?
>
> Just like in MuZero (Schrittwieser et al., 2019) and related works, the latent dynamics model in GLAMOR is not required to model all the information in the original observation. Instead, the latent states are only incentivized to contain information necessary to predict the actions that are necessary to take the agent to the goal state. We are not aware of any theoretical work about the relationship between inverse dynamics and controllability, but this claim is also made in ICM (Pathak et al., 2017).
>
> Q: And have any empirical studies been done to show that this actually helps model-based RL to perform better than when using forward models? It would be interesting to compare against model-based RL techniques based on forward models, if available, for this work.
>
> One of the main benefits of using an inverse dynamics model is that we can use the factored probabilities to guide the search for a strong action sequence. This is in contrast to how an algorithm like MuZero (Schrittwieser et al., 2019) models the expected reward at each time-step and learns a separate policy/value function to guide search. We added a new experiment in our updated PDF to test the extent to which using an inverse dynamics model helps with planning versus directly modeling the reward the agent cares about (probability of reaching the goal) and planning using just sparse information.
>
> We used RS in this experiment as well, sampling action sequences at random and picking the sequence with the highest score out of the sampled sequences. We found that in our environments, this agent performed significantly worse with the same level of compute, showing the value added by learning the inverse dynamics model.
>
> We leave further comparisons with other MBRL techniques to future work.

---

### Official Review · AnonReviewer4 · 2020-10-30
**Review for Planning from Pixels using Inverse Dynamics Models**

**Rating:** 6
**Confidence:** 3

**Review:**

Summary: The author proposes Goal-Conditioned Latent Action Models for RL (GLAMOR) a novel approach to learn latent world models by modeling inverse dynamics. The proposed approach learns to track task-relevant dynamics for a diverse distribution of tasks and provide a strong heuristic that enables efficient planning. GLAMOR demonstrates good performance against its baselines in terms of achieving accurately goals, sample efficiency and effective planning.


Reason for the score:
Paper address an interesting question related to Reinforcement Learning and provides strong results to back the proposed method.


Pros:
-good flow of the paper

-Correct situation of the problem with respect to current research

-A good problem to address in Reinforcement Learning area

-Strong experimental results


Cons:
-Can the authors talk about how the proposed method perform in a real-robotic task?

-In the equation 5, a_{<i} means?

-in page 3 last para is not clear to me. Can you rewrite or provide explanation of how it is possible?

-In para 3 of Section 3.4, why you sample from Boltzmann distribution and not uniform distribution?

-Have you considered other exploration methods than of epsilon-greedy?


Questions for the rebuttal:
Please address and clarify the cons above

---

> ### Author Response · Authors · 2020-11-17
> **Thank you for the comments**
>
> Thank you for the valuable comments. We have uploaded a new PDF. Please see general comments to all reviewers for details about what has changed.
>
> Q: Can the authors talk about how the proposed method performs in a real-robotic task?
>
> We think that our method is a good fit for real-robot tasks, due to its especially strong performance with off-policy data (appendix A.4.1) and the fact that the goal-achievement task probably won’t require expensive env.reset() operations, which are usually a pain-point in real-robot experiments. These experiments still require expensive equipment and significant time, so we leave such experimentation to future work.
>
> Q: Confused about what  a_{<i} means:
>
> We have clarified the notation in our updated PDF. This just means a_1, …, a_{i-1}.
>
> Q: On page 3 the last para is not clear to me. Can you rewrite or provide an explanation of how it is possible?
>
> We apologize that this was not clear. We have rewritten the paragraph (now on page 4). The main point is that, with our objective factored over the individual actions, we can use this information to guide our search. It is similar to in NLP, where if we wanted to find a sequence of tokens that have a high probability under a language model without using any intermediate information, it would be quite challenging. Since we have access to the intermediate p(token_i|token_{<i}) however, the search becomes a lot easier.
>
> Q: In para 3 of Section 3.4, why do you sample from a Boltzmann distribution and not uniform distribution?
>
> This is just one way to guide the search based on the factored probabilities. As in the example above, using a uniform distribution to search for action sequences doesn’t use this information and won’t result in a very good action sequence (see section 4.7.3 in the updated PDF, where we added an additional experiment trying exactly this). When we sample using the z_i, almost every action sequence sampled gets the agent to the goal state, so the search becomes easy.
>
> Q: Have you considered other exploration methods than epsilon-greedy?
>
> We use epsilon-greedy simply to guarantee that each goal has at least some probability of being reached (needed for our derivation). As mentioned in the new section 3.6, GLAMOR’s strong performance with off-policy data suggests that more sophisticated exploration schemes could probably be used. However, we found that epsilon-greedy was sufficient, so we did not experiment with different methods. We consider this direction to be orthogonal to the main contribution of this paper. Future extensions could include different ways to sample goals that lead to more efficient exploration, or ways to choose actions specifically to gain the most information about the parameters of the inverse dynamics model. This is indeed an interesting area for future work.

---

### Official Review · AnonReviewer3 · 2020-11-03
**Interesting work. Need more justification on the main claim**

**Rating:** 6
**Confidence:** 4

**Review:**

The paper proposes a model-based reinforcement learning method. The method builds a partial model of the environment through learning inverse dynamics, which is the distribution of action sequences that would bring one state to another state. Through training the model with an iterative relabeling scheme, the model is able to learn to reach goals in a subset of DM Control and Atari domains.

Comments:
+ This MBRL formulation is conceptually simple and does not rely on learning the full model of an environment, potentially enabling long-horizon planning while being able to high-dimensional input.
+ The relationship and differences to the closely related work Ghosh et al., 2020 is clearly stated and elaborated.

- My main concern about the paper is that although the difference between GCSL and the proposed method is exposited in theory (Proposition 3.1), there is no direct empirical experiment explaining how this difference might play out in practice. Specifically, there clearly are performance gaps between the proposed method and GCSL, but I cannot seem to grasp what EXACTLY is causing this gap. Is it ONLY because of the difference stated in proposition 3.1? If so, I'd like to see a minimum toy experiment clearly showing the qualitative differences. If there are other differences, I'd like to see them addressed in greater details. None of the reasons (i)-(iii) stated in section 3.6 are substantiated experimentally.

---

> ### Author Response · Authors · 2020-11-17
> **Thank you for the comments**
>
> Thank you for the valuable comments.
>
> We agree that it was unclear exactly what was contributing to the gap in performance between GLAMOR and baselines. To answer your questions, we have uploaded a new PDF (please also see general comments to all reviewers for more details about what has changed) with many extra experiments. What follows is a summary of the new experiments and results:
>
> 1. We ran an experiment in Appendix A.4.1 that shows that if we simply sample from the inverse dynamics model in GLAMOR during training (essentially the same as just sampling one action sequence in our planner), the performance is much worse than when it plans (samples many sequences and selects the one with the highest score).
> 2. Also in Appendix A.4.1, we also show that without using the action prior, GLAMOR performs significantly worse.
> 3. We also show that GLAMOR performs significantly better when training data produced by a random agent. While GCSL relies on iterates of “gather training data from the current policy” and “train on the relabeled data”, our algorithm has no such requirement. In fact, data collected from a uniformly random policy is great for GLAMOR since the action prior will be trivial.
> 4. GLAMOR has the ability to predict that it is currently at the goal state and terminate the episode and evaluate early while baselines always evaluate whether the goal was achieved after a fixed number of steps. In Section 4.7.2, we evaluated the effects of early termination on the performance of the agent to see whether the performance gain could be attributed to this. We found that early termination strategy in GLAMOR is not necessary if the planner explicitly plans to reach the goal at the last time-step in the trajectory. This also illustrates a benefit of GLAMOR: it learns a world model rather than a policy, which makes it significantly more flexible.
> 5. One of the main benefits of using an inverse dynamics model is that we can use the factored probabilities to guide the search for a strong action sequence. This is in contrast to how an algorithm like MuZero (Schrittwieser et al., 2019) models the expected reward at each time-step and learns a separate policy/value function to guide search. We added a new experiment in section 4.7.3 to test the extent to which using an inverse dynamics model helps with planning versus directly modeling the reward the agent cares about (probability of reaching the goal) and planning using just sparse information.
>
>   We used RS in this experiment as well, sampling action sequences at random and picking the sequence with the highest score out of the sampled sequences. We found that in our environments, this agent performed significantly worse with the same level of compute, showing the value added by learning an inverse dynamics model.
>
> We believe these results provide better insight into where the differences in performance come from.

---

### Author Response · Authors · 2020-11-17
**General Comments to All Reviewers**

We thank all the reviewers for the valuable comments. We agree with the common sentiment that while the differences between our method and the baselines are clear in terms of performance, our experiments didn’t do enough to explain exactly why this difference exists. We have uploaded a new version of our PDF that has several changes and new experiments that hope to address these concerns. These changes are:

1. Added an ablation study in Appendix A.4. This section includes an experiment testing the effect of using the action prior (without the action prior, performance is worse) and without planning (without planning, performance is significantly worse). We also tested GLAMOR and GCSL trained on a dataset collected with a random policy. GLAMOR performs almost as well as the full algorithm while GCSL sees a large decrease in performance (this was also shown in their original paper).
2. In section 4.7.3, we tested the agent’s performance when we restricted the planner to only use sparse information rather than use the intermediate probabilities provided by the inverse dynamics and action prior. This resulted in much lower performance with equal levels of search compute.
3. We added a new experiment in section 4.7.2 that shows that our GLAMOR agent can adjust its behavior to reach a goal at a specific time-step, achieving nearly the same performance as the version which can terminate an episode early. The agent’s world model has knowledge about many different action sequences that lead to the goal - a benefit over the baselines. This also shows that the performance improvement is not mainly due to differences in evaluation.
4. We moved the old section 3.6 (GCSL) to the appendix and replaced it with a new “Comparison to Prior Methods” section. This section highlights that GLAMOR: learns a world model that is more flexible than a simple policy or Q-network; GLAMOR performs well with off-policy data; and GLAMOR uses an action prior to avoid the interference problem in GCSL.
5. We added Appendix A.3 with hyperparameters and implementation details. We also added a figure illustrating the network architecture (Figure 1).
6. We added a link to an anonymous website with videos of the GLAMOR agent reaching goals in all tested environments as well as videos of the termination strategy experiments.
7. (New 11/23) We updated the styling of the plots to improve visibility.

---

### Decision · Program_Chairs · 2021-01-07
**Final Decision**

**Decision:**

Accept (Poster)

**Comment:**

While many work in the literature (PlaNet (2018), Dreamer (2020), SimPLe (2019), etc.) learn world models to perform well on a particular task at hand, the motivation behind this work is that dynamics models benefit if they are task-agnostic, hence would be able to perform a wider range of tasks, as opposed to just doing one task really well. In order to do this, they propose to learn a latent representation that models inverse dynamics of the system / environment rather than capturing information about the task-specific rewards, and incorporate a planning for solving specific tasks in which they can measure performance.

To show broad applicability of their method, the authors tested their approach on Atari and DM Control Suite (from pixels), and also simple grid worlds to illustrate the concepts, and demonstrated strong performance over SOTA model-free algorithms (even the ones that do not have open-source implementations). Reviewers and myself agree that the paper is well written, easy to follow, and the approach is well-motivated.

After the review period, the authors have done work to improve the draft, particularly including ablation studies with and without planning, addition comparisons, and improved visualizations, after taking in the comments and feedback from the reviewers after the initial reviews, which satisfied some of the reviewers. One reviewer asked for a real robotic task, but I feel that while it will help the paper, many existing works focus purely on DM control from pixels, and this work has performed experiments on both DM Control and Atari, two reasonably different domains, and IMO makes up for the lack of real-world robotics experiment. That being said, a discussion on how the proposed method would work in a real-robotic task, as suggested by R4 would be good to have.

I believe the work in its current state is ready for acceptance for ICLR 2021, and should be a fine contribution to the visual model-based RL works. I'm excited to see this work presented to the community, and I'm going to recommend acceptance (Poster).